# Application of Deep Neural Networks as a Prescreening Tool to Assign Individualized Absorption Models in Pharmacokinetic Analysis

**DOI:** 10.3390/pharmaceutics13060797

**Published:** 2021-05-26

**Authors:** Mutaz M. Jaber, Burhaneddin Yaman, Kyriakie Sarafoglou, Richard C. Brundage

**Affiliations:** 1Department of Experimental and Clinical Pharmacology, College of Pharmacy, University of Minnesota, Minneapolis, MN 55455, USA; jaber038@umn.edu (M.M.J.); saraf010@umn.edu (K.S.); 2Department of Electrical and Computer Engineering, University of Minnesota, Minneapolis, MN 55455, USA; yaman013@umn.edu; 3Department of Pediatrics, University of Minnesota, Minneapolis, MN 55454, USA

**Keywords:** pharmacokinetics, deep learning, machine learning, absorption models, visual inspection, individualized models

## Abstract

A specific model for drug absorption is necessarily assumed in pharmacokinetic (PK) analyses following extravascular dosing. Unfortunately, an inappropriate absorption model may force other model parameters to be poorly estimated. An added complexity arises in population PK analyses when different individuals appear to have different absorption patterns. The aim of this study is to demonstrate that a deep neural network (DNN) can be used to prescreen data and assign an individualized absorption model consistent with either a first-order, Erlang, or split-peak process. Ten thousand profiles were simulated for each of the three aforementioned shapes and used for training the DNN algorithm with a 30% hold-out validation set. During the training phase, a 99.7% accuracy was attained, with 99.4% accuracy during in the validation process. In testing the algorithm classification performance with external patient data, a 93.7% accuracy was reached. This algorithm was developed to prescreen individual data and assign a particular absorption model prior to a population PK analysis. We envision it being used as an efficient prescreening tool in other situations that involve a model component that appears to be variable across subjects. It has the potential to reduce the time needed to perform a manual visual assignment and eliminate inter-assessor variability and bias in assigning a sub-model.

## 1. Introduction

In population pharmacokinetic/pharmacodynamic (PK/PD) modeling, specific structural models are chosen to characterize the shape of the observed concentrations or effects vs. time. These structures are usually obvious and common to all subjects. However, sometimes additional consideration needs to be given to the absorption process from the depot compartment to the central compartment. Given the biological complexity that underlies absorption models and the inherent variability associated with the processes across individuals, it is not surprising that data from different individuals in a dataset may need different models to describe the data. In population analyses, two approaches are frequently used. We might completely ignore the model misspecification that exists in some individuals and allow the inflation of between-subject variability (BSV) and residual unexplained variability (RUV) to accommodate the misspecification; or we might assume a more highly complex model and borrow parameter information from those subjects who are able to support the complexity of the model while letting others shrink toward the typical values of the estimates. When observations taken during the period of drug absorption are sparse, the choice of absorption model or even their specific parameter values are likely trivial and have little impact on the remaining parameter estimates [1]. In addition, the relative sparse sampling during absorption generally results in absorption parameter estimates with high imprecision (large relative standard errors) and frequently large BSV. However, as the frequency of observations taken during absorption increases, the complexities of drug absorption become apparent and obvious misspecification can sometimes be seen in diagnostic plots. Although the absorption sub-model can be adapted to provide a better fit to the data, the increased complexity of the model will then require an increased frequency of sampling at critical times to capture those absorption parameters with acceptable precision. A further complicating consideration can occur in population PK analyses as the intensity of sampling during absorption increases. Some subjects might clearly demonstrate one absorption pattern while others a different pattern. Nonetheless, a single absorption model is generally assumed. Examples of more involved techniques such as mechanistic models [2], or mixture modeling [3] have been applied in these situations.

Machine learning is gaining wider attention in clinical pharmacology as computational capacity increases. Methods for machine learning use statistical algorithms that are capable of doing automated learning from existing data to uncover patterns [4,5,6,7,8]. Deep learning is a branch of machine learning that involve artificial neural networks in their structure to facilitate developing algorithms capable of learning from data [5,6]. Neural networks are not new to clinical pharmacology and have been applied in response classification, dose selection, and quantitative system pharmacology model reduction [9,10,11].

We have recent experience in a population PK analysis in which the absorption process demonstrated concentration-time profiles that, while not apparent on standard mean or spaghetti plots, appeared on closer examination to have individuals that conform to either a first-order, Erlang, or a split-peak process, depending on the individual. Our initial approach was to fit a Erlang-distribution absorption model to the data [12]. Given that the specifics of the absorption process weren’t of primary importance, the input process was considered a trivial component of the analysis. Subsequently we explored visually prescreening the individual profiles to make an assignment of the absorption model prior to modeling the data [13]. It was an interesting exercise, but we found this procedure to be less than satisfactory as visually assessing the profiles took considerable time and demonstrated inter-rater variability. The exercise motivated us to seek an alternative approach in assigning the absorption model. The goal of this study was to build a deep neural network (DNN) algorithm to recognize these absorption profiles and apply it to our data as a means to evaluate the performance of the method in assigning the absorption model structure for each individual.

## 2. Methods

### 2.1. Observed Data and Visual Assignment

The data used for this study are previously described in a nonlinear mixed-effects analysis of cortisol [12]. Briefly, after the study was approved by the University of Minnesota Institutional Review Board (Project 1209M21101 approved on 5 July 2017). Following each subject’s morning dose at 0800, 12 concentrations were obtained at times 0 (Predose), 0.25, 0.5, 0.75, 1, 1.25, 1.5, 2, 2.5, 3, 4, and 6 hours after the dose. Concentration-time data consisting of 682 cortisol observations from 53 patients were available. Post-publication, we visually assessed the concentration-time data and assigned each individual as having either a first-order absorption process (n = 20), absorption consistent with an Erlang distributed delay model (n = 21), or a split-peak absorption process (mixed first-order, and Erlang models; n = 12) [13] These absorption patterns are exemplified in Figure 1.

### 2.2. Estimation of Pharmacokinetic Model

With these individualized absorption models assigned, we performed an additional population PK re-analysis that estimated the parameters of the pre-specified absorption models in addition to clearance (CL) and the volume of distribution (V). NONMEM 7.5 (ICON plc development LLC) using first-order estimation with interaction (FOCE-I) was used. Figure 2 illustrates the pharmacokinetic structural models of the three absorption profiles. Table 1 presents the final parameter estimates from the re-analysis. The NONMEM control stream, and individual profiles of observed and predicted concentrations as linear (Appendix A) and semi-log (Appendix A) are available in Appendix A.

### 2.3. Simulation of Training Profiles

The parameters from the above analysis were used to generate simulated data sets for training the DNN. Between-subject and residual unexplained variability random effects were included in the simulation to assure a different profile for each simulated individual. For the purpose of training the DNN, simulations were based on a standard 20-kg subject with a 10-mg dose being administered. Concentrations were simulated at the same 12 time points as the external dataset. Ten thousand profiles were simulated for each of the three distinct absorption models (first-order, Erlang, and split-peak shapes) with CL and V shared among the three absorption models. The R package mrgsolve v0.10.7 used for the simulations [14].

These simulated concentration-time profiles were then standardized using the Feature scaling method [15], in Equation (Equation 1) to standardize all concentrations between zero and one while maintaining the shape of the profile.
(1)$$CS_{i,j}=\frac{C_{i,j}-C_{min,j}}{C_{max,j}-C_{min,j}}$$
where $CS_{i,j}$ is the $i^{th}$ standardized concentration for the $j^{th}$ individual, $C_{i,j}$ is the $i^{th}$ simulated concentration for the $j^{th}$ individual, $C_{min,j}$ is the minimum concentration for the $j^{th}$ individual, and $C_{max,j}$ is the maximum concentration for the $j^{th}$ individual. From this simulated dataset of 30,000 standardized profiles, 30% (10% from each absorption model) were randomly selected as hold-out validation data to evaluate the performance of the algorithm during the training.

### 2.4. Deep Learning Algorithm

The open-source R library packages TensorFlow v2.2 and Keras v2.0 were used to develop the DNN algorithm. A DNN consists of an input layer, hidden layers, and an output layer. Each hidden layer consists of a number of nodes that represents the computational unit. The output from each node in a layer will propagate as input to each node of the subsequent layer. More formally, the layers of a DNN and a diagram of one node is presented in Figure 3. Equation (Equation 2) presents the output of a given node.
(2)$$y_{n,l}=f\left(b_{l}+\sum_{n=1}^{N}x_{n}w_{n,l-1}\right)$$
where $y_{n,l}$ is the output value of the nth node in the $l^{th}$ layer. $b_{l}$ is the bias in the $l^{th}$ layer, *N* represents the number of nodes in layer l−1 (the previous layer), $x_{n}$ is the value of $n^{th}$ node that is being propagated forward from layer l−1, and $w_{n,l-1}$ is the weight associated with the $n^{th}$ input from the l−1 layer to the $n^{th}$ node in the $l^{th}$ layer. f(.) is the activation function defined in the algorithm (vide infra).

In our case, the input layer consisted of the 12 Feature-scaled simulated concentrations ($CS_{i,j}$) at the aforementioned sampling times for each of the simulated subjects. The number of hidden layers was sequentially tested from 1 to 6 and evaluated using the resulting accuracy and loss function value (vide infra). The final output from the DNN was simply the probability that the data set was represented by each of the three absorption profiles (first-order, Erlang, or split-peak).

ReLU was used as a linear activation function at the hidden layers for each node and the softmax activation was used in the last layer (output) for the purpose of classification [16]. Categorical cross-entropy was defined as the loss function [17] with a batch size of 32 over 100 epochs.

### 2.5. Evaluation

The “true” absorption profile shape was taken to be the shape decided by visual inspection in the estimation process and used for the accuracy determinations. At the end of the training phase, the overall categorical cross-entropy loss function value for both the training data split (70%) and the validation data split (30%) was calculated and the accuracy of the algorithm was evaluated using the number of correct predictions over the total number of simulated profiles (training and validation) in the dataset [18]. The accuracy selection is based on the index of the highest probability in a vector of three indexes that represent the probabilities of the three absorption shapes. Hence, if the predicted and observed shape match, it will count as an accurate prediction.

Of the 53 external subject profiles, 5 profiles were not complete datasets and 48 profiles from the PK study were used as external data to evaluate the algorithm classification performance as a prescreening tool. The percentages of correct classifications were calculated using a confusion matrix that summarizes the performance of algorithm prediction in comparison to the “true” absorption profile. In addition, the accuracy rate was compared to the uninformative rate (correct classification due to chance) of the external data to determine the extent to which the algorithm chose the correct shape using a binomial test with p<0.05 regarded as significant.

## 3. Results

No improvement was noted in the accuracy or in the loss function value when increasing the number of hidden layers beyond three and all results are shown for three hidden layers. Table 2 summarizes the number of profiles used in training, validation and external datasets with corresponding overall accuracy and the values of loss function.

Figure 4 presents the output from three representative profiles (first-order, Erlang, and split-peak) demonstrating the probabilities of each absorption profile.

Classification results are presented in Table 3 as a confusion matrix and contains the percentages of predicting the correct shape on the diagonal and the percentage of choosing the incorrect model on the off diagonals.

A difference between the accuracy rate of 93.7% and the uninformative rate of 45.8% was significant (p<0.001).

Of the 45 correct decisions, all were above the probability of 0.75; 89% exceeded a probability of 0.9 (N = 40). For the three incorrect decisions, the probabilities were 0.53, 0.50, 0.56, while the true shape probabilities were 0.47, 0.48, and 0.42, respectively. Figure 5 displays a histogram of the probabilities of the classifications for all 48 subjects.

## 4. Discussion

The suggested DNN algorithm can be used in the pre-modeling setting but it must be recognized that the approach might be most appropriate when the precision of a sub-model is perhaps of little consequence as with absorption. The raison d’être of the method can be stated as an approach to minimize the potential of having misspecification in one sub-model adversely affect the estimation of parameters in another part of the model, say, clearance or volume of distribution.

It is important to note that our visual labeling of a profile as the “true shape” does not make it truth. A “true shape” does not exist in real data. It is encouraging that when the DNN algorithm is not in agreement with the original visual classification, the certainty of that decision is low as evidenced by lower probabilities. Indeed, when the DNN probabilities are relatively balanced across outcomes as we observed with incorrect classifications, the choice of pattern is likely inconsequential.

Mixture models are an alternative approach in addressing these issues. It assumes that a given parameter distribution can be composed of 2 or more subpopulations and the software will compute the probability of being in each subpopulation and classify each individual to the subpopulation that is most probable [3]. Although applying a mixture model to absorption may minimize misspecification of another part of the PK model, this approach will likely consume several degrees of freedom during the estimation step. Additionally, it is noteworthy that the computational time is increased. To our knowledge, while mixture models are frequently used to better understand multimodal parameter distributions, they have not yet been applied to classify subjects into having one of several competing absorption models.

Knowledge of the steps and processes involved with pharmacokinetics are becoming better understood and the complexities of absorption are captured by physiologically-based mechanistic models that have been adapted to explain differences in absorption profiles [19]. However, due to the intensive demand of these models for prior information, approximate mechanistic models have been developed. A gastro-intestinal transit time model (GITT) [2] was used to characterize the timing of tablet movement in the intestine using a step function based on prior information, and with the help of mixture modeling, different absorption rates in different GI regions were estimated. Ruiz-Garcia et al. [20] observed non-standard absorption profiles for dacomitinib with and without proton pump inhibitors. They evaluated a series of increasingly complex absorption models before assuming a global transit compartment model. As is the usual case, it does not appear that they attempted to allow more than one absorption model across the subjects. A more complicated method to describe absorption process has been suggested by Csajka et al. [21] where they described the absorption after the administration of hydromorphone and veralipride by the sum of inverse Gaussian functions. Although empirical, this method was able to describe complicated absorption shapes such as the double peak phenomena with much more flexibility than simpler ones, was also able to characterize BSV. A more complicated empirical method using fractional-order kinetics have been suggested to describe anomalous absorption kinetics based upon drug dissolution processes [22].

With all the simplifications that are often imposed on the absorption process, it is not always trivial. The early exposure of drug is certainly affected by drug absorption characteristics and has been shown to be important since it is associated with the onset of response and clinical outcomes. This has led the US Food and Drug Administration to issue draft guidances for methylphenidate [23], hydromorphone [24], and amantadine [25]. This underscores the potential importance of the absorption process, particularly in the development of generic formulations.

A limitation of this report is that the current DNN algorithm has been developed and trained using parameters from a cortisol population-based PK analysis, and then using that same data as the external data set for qualification. A more fair assessment would of course be to use an independent data set, but one wasn’t available. The purpose of this report is to demonstrate the potential utility of using DNN to classify model components. It has not been applied to another situation and it might be the case that the DNN will need to be trained uniquely with data relevant to each setting. It is also recognized that while the training sets were balanced across the shapes, the external dataset were not. We acknowledge that the approach may perform differently with drugs having different PK characteristics. DNN algorithms are by nature data-driven and this approach can be used in conjunction with pharmacometrics analyses to improve model building and estimation process. In the future, it may be possible to build these approximation functions directly into the estimation process.

## 5. Conclusions

In summary, the developed DNN algorithm was capable of predicting different absorption profiles in a population with high accuracy in both simulated and external datasets. This DNN pre-specification algorithm may reduce the computational time of a mixture model analysis and avoid consuming unnecessary degrees of freedom during estimation and may obviate the need to obtain or generate compound-specific prior information for complex physiological absorption processes. Finally, this algorithm will reduce valid concerns of inter-assessor variability and bias when visually assigning the absorption shapes.

## Figures and Tables

**Figure 1 pharmaceutics-13-00797-f001:**
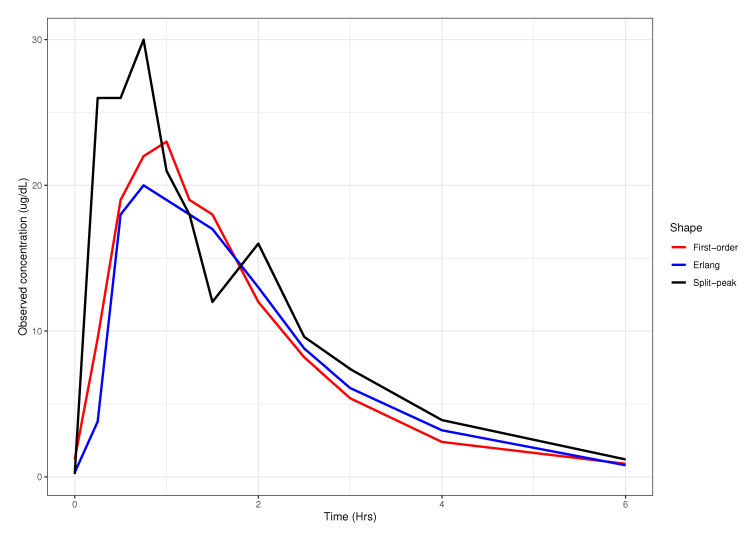
Observed concentration-time profiles for three representative shapes. (red) presents the first-order process; (blue) presents the Erlang process; (black) presents the mixed first-order and Erlang processes.

**Figure 2 pharmaceutics-13-00797-f002:**
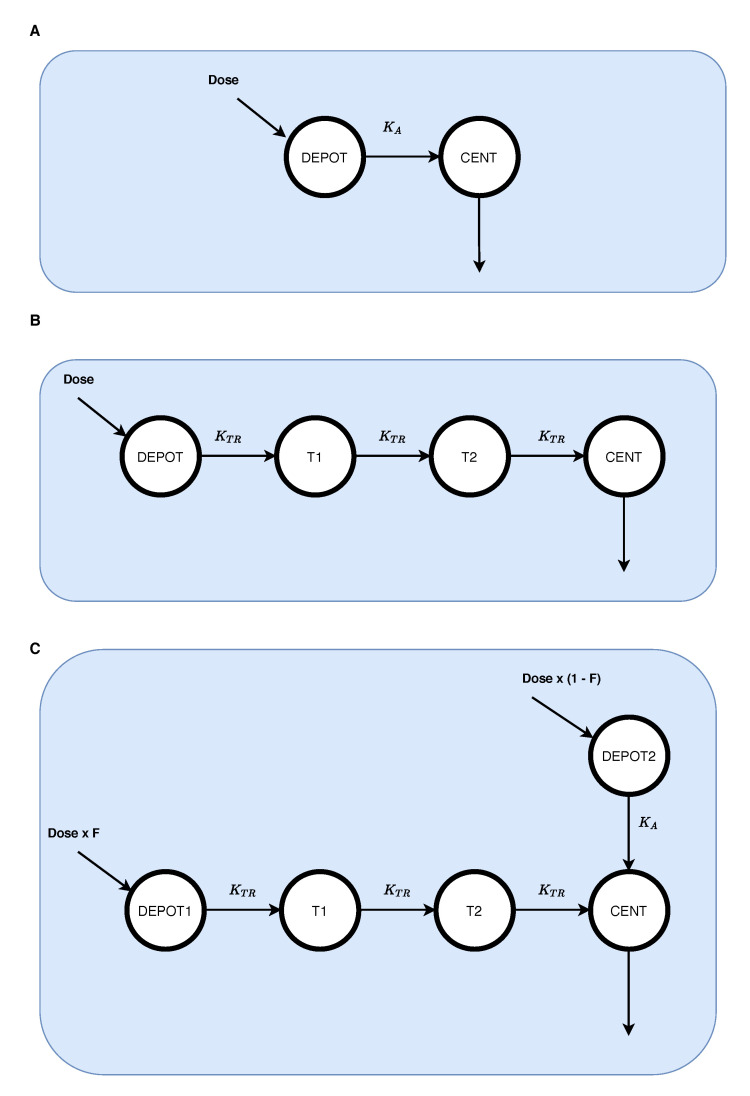
Pharmacokinetic structural models of the three absorption profiles. (**A**) First-order absorption; (**B**) Erlang absorption process; (**C**) Mixed first-order and Erlang absorption.

**Figure 3 pharmaceutics-13-00797-f003:**
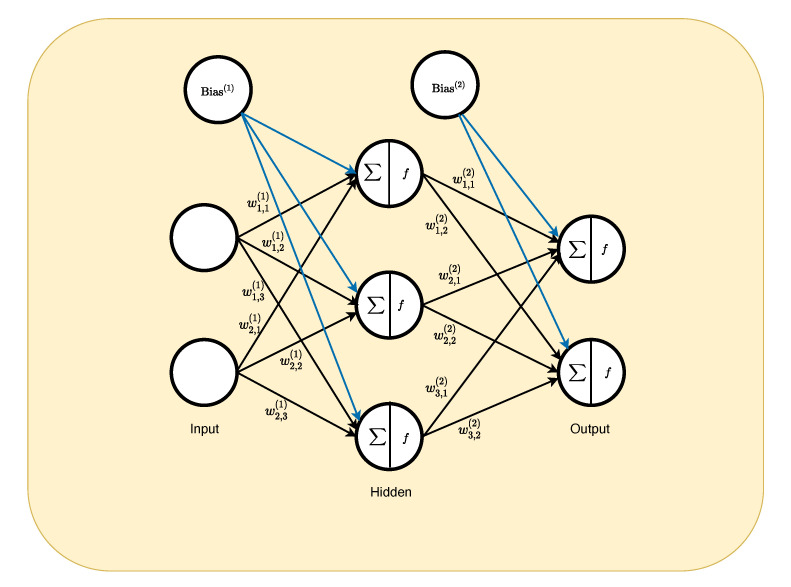
Overview of deep neural network algorithm structure.

**Figure 4 pharmaceutics-13-00797-f004:**
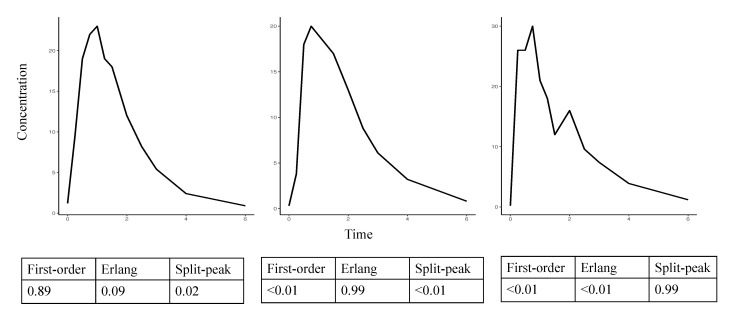
The output of the developed DNN with three examples from the external cortisol data.

**Figure 5 pharmaceutics-13-00797-f005:**
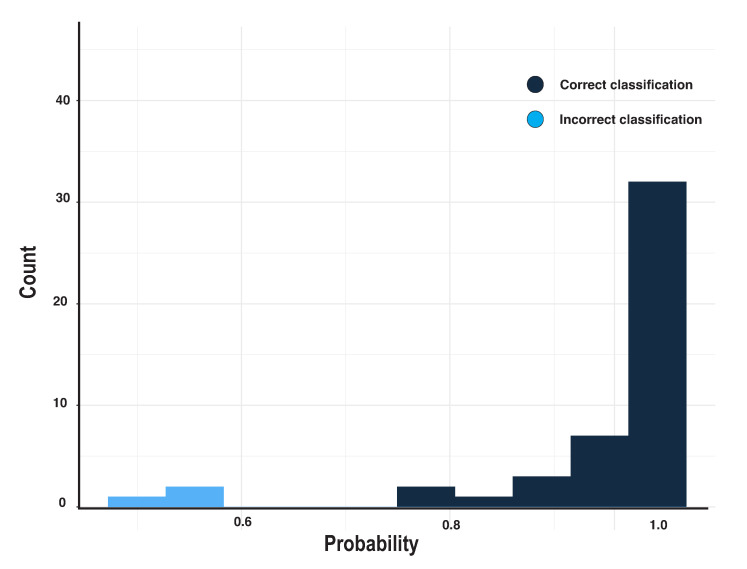
Probability counts for DNN model prediction. Dark blue presents the correct classification; Light blue presents the incorrect classification.

**Table 1 pharmaceutics-13-00797-t001:** Population-level pharmacokinetic estimates and the between-subject variabilities (% CV) from the reanalysis used in the simulation.

Parameter	First-Order	Erlang	Split-Peak
CL (L/h/70 kg)		22.6 (29%)	
V (L/70 kg)		38.9 (21%	
$K_{\mathrm{TR}}$ ($h^{-1}$)	-	8.2 (23%)	5.2 (23%)
$K_{A}$ ($h^{-1}$)	3.3 (48%)	-	7.6 (48%)
Fraction (%)	-	-	78 (80%)
${\mathrm{RUV}}_{\mathrm{prop}}$		16.5%	

**Table 2 pharmaceutics-13-00797-t002:** Collection of data with associated overall accuracy and loss value.

Data	N	Overall Accuracy	Overall Loss Value
Training	21,000	99.7%	<0.01
Validation	9000	99.4%	<0.01
External	48	93.7%	0.17

**Table 3 pharmaceutics-13-00797-t003:** Confusion matrix presents the classification of the external patient data.

DNN Prediction	Visual Assignment		
	First-Order	Erlang	Split-Peak
First-order	18	1	1
Erlang	0	21	1
Split-peak	0	0	6

## Data Availability

Data sharing not applicable.

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
