# Peer review of "Application of Deep Neural Networks as a Prescreening Tool to Assign Individualized Absorption Models in Pharmacokinetic Analysis"

_pharmaceutics, 2021, doi:10.3390/pharmaceutics13060797_

Round 1
Reviewer 1 Report
The manuscript aimed to use DNN as a prescreening tool to assign individualized absorption models in pharmacokinetic analysis. This is a novel and interesting topic.
Just one question needs to be justified. The author used ten thousand simulated PK profiles for DNN training and validation. Please give more explanation and justification of the methodology.
8 shapes and used for training the DNN algorithm
Reviewer 2 Report
The authors have shown the role of deep neural network algorithm as a tool to optimize the selection of the absorption model at individual level. The article is well-written and provides new insights on how artificial intelegence could help the decision-making process during the drug development. Several concerns have been identified that need to be solved by the athors:
- Figure 1 depicts the PK profile of the three absorption models considered. However, it would be informative to provide the individual predictions of each absorption model together with the AIC values in order to understand the selection of the optimal absorption model. It could be provided as supplementary material
- The authors state that different absorption models could be selected at the individual level in order to characterize the time-course profile of the drug. However, mechanistic models (PBPK) may contribute to understanding regional differences during the drug dissolution, transit, and absorption within the GI rather than assuming different structural absorption models. In this sense, the authors should clarify whether the conclusions of the current work are highly dependent on the data available or more complex absorption models (PBPK) would lead to the selection of the same model across all individuals.
- Individual experimental profiles should be incorporated in the manuscript to understand the number of samples and sampling strategy of each individual.
- The authors should clarify whether the learning dataset and external dataset were balanced across the different absorption models or whether different accuracy results would be expected when the different proportions of absorption behaviors are selected.
Reviewer 3 Report
The study seems interesting but it is important to provide more details on the construction of the DNN model as well as to make available the R scripts used
Round 2
Reviewer 3 Report
I have no other comments!